# The Potential Role of Migratory Birds in the Rapid Spread of Ticks and Tick-Borne Pathogens in the Changing Climatic and Environmental Conditions in Europe

**DOI:** 10.3390/ijerph17062117

**Published:** 2020-03-23

**Authors:** Alicja M. Buczek, Weronika Buczek, Alicja Buczek, Katarzyna Bartosik

**Affiliations:** Chair and Department of Biology and Parasitology, Medical University of Lublin, Radziwiłłowska 11 St., 20-080 Lublin, Poland; abuczek21@gmail.com (A.M.B.); wera1301@gmail.com (W.B.); katarzyna.bartosik@umlub.pl (K.B.)

**Keywords:** climate change, environmental changes, behaviour of migratory birds, ticks, zoonoses, tick-borne diseases

## Abstract

This opinion piece highlights the role of migratory birds in the spread of ticks and their role in the circulation and dissemination of pathogens in Europe. Birds with different lifestyles, i.e., non-migrants residing in a specific area, or short-, medium-, and long-distance migrants, migrating within one or several distant geographical regions are carriers of a number of ticks and tick-borne pathogens. During seasonal migrations, birds that cover long distances over a short time and stay temporarily in different habitats can introduce tick and pathogen species in areas where they have never occurred. An increase in the geographical range of ticks as well as the global climate changes affecting the pathogens, vectors, and their hosts increase the incidence and the spread of emerging tick-borne diseases worldwide. Tick infestations of birds varied between regions depends on the rhythms of tick seasonal activity and the bird migration rhythms determined by for example, climatic and environmental factors. In areas north of latitude ca. 58°N, immature *Ixodes ricinus* ticks are collected from birds most frequently, whereas ticks from the *Hyalomma marginatum* group dominate in areas below 42°N. We concluded that the prognosis of hazards posed by tick-borne pathogens should take into account changes in the migration of birds, hosts of many epidemiologically important tick species.

## 1. Introduction

Ixodid tick species in Europe represent five genera: *Ixodes*, *Dermacentor*, *Haemaphysalis*, *Hyalomma*, and *Rhipicephalus*, some of which have great veterinary and medical importance [1]. Their spread is mainly determined by climatic and environmental conditions and the presence of animals, i.e., potential hosts of all tick developmental stages. Due to global warming and other effects of anthropopressure, e.g., changes in the fauna and flora structure and ecological fragmentation that became strongly evident in the 20th century and at the beginning of the 21st century, the distribution of ticks is changing, as these are colonising new areas. In this period, the area of occurrence of such pathogen-transmitting tick species as *Ixodes ricinus* [2,3,4], *Dermacentor reticulatus* [5,6,7], *Dermacentor marginatus* [7], *Haemaphysalis concinna* [8], *Hyalomma marginatum* [9,10], and *Rhipicephalus sanguineus* [11] in Europe increased significantly. Vector competences of these ticks for specific pathogen species vary, but they are all probably involved in pathogen circulation in nature and contribute to the maintenance of tick-borne disease foci. 

The genera *Dermacentor* and *Rhipicephalus* usually do not parasitize birds, as their larvae are only sporadically found on these hosts [12,13,14]. Tick feeding which consists of the introduction of saliva into the host organism alternately with blood meal uptake, may lead to transmission of pathogens in both directions: from the tick to the host and vice versa. Infected ticks may transmit pathogens in the host’s blood vessel that may be picked up by other ticks feeding in the vicinity on the same host’s body at the same time [15]. The transfer of certain pathogenic and non-pathogenic microorganisms can also take place via conspecific and interspecific tick parasitism [16] or, probably, during oral-anal contact between two different tick species–*I. ricinus* and *D. reticulatus* [17].

In natural conditions, ticks move over short distances. *Ixodes scapularis* nymphs and adults, cover a distance of only 2–3 m and 5 m, respectively [18]. Within 7 weeks, adult *D. reticulatus* ticks can cover an average distance of 60.71 ± 44 cm [19]. Ticks are transmitted from one habitat to another mainly by avian and mammalian hosts [20,21,22,23]. These hosts migrate regularly (cyclically), which is related to e.g., reproduction cycles and recurrent changes in the environment. In turn, their irregular migrations are most often related to adverse environmental conditions (e.g., lack of feed or water) and overpopulation. Ticks infesting the skin of their avian and mammalian hosts are spread within and among habitats. Ticks attached to mammalian fur can also be transferred over certain distances. We found labelled unengorged adult *D. reticulatus* ticks at a distance of 2 to 3 km away from the site where they were released. They were probably transferred on mammalian fur or on the clothes of forest workers that were present in the habitats of these ticks [19], (Bzowski, unpublished data). In favourable conditions, ticks can colonise a new habitat and reproduce successfully. However, determination of the impact of birds on the transmission and fauna of ticks in various regions requires further research, including investigations of molecular ecology with genetic methods based on genetic markers or radioactive isotopes.

Each hard tick stage feeds one time, and most species in Europe feed on three different hosts sequentially, but some species feed only on two hosts as two developmental stages will feed on the same host. The change of hosts during the life cycle promotes the circulation of pathogens, which are introduced by active tick stages with saliva during feeding. The wide spectrum of tick hosts with great epidemiological importance promotes quick spread and long-term persistence of tick-borne pathogens in the environment.

Ticks can be transported by birds within Eurasia and between Eurasia and Africa. This is accompanied by spread of tick-borne zoonotic pathogens [12,24,25,26,27]. The length of tick foraging varies from a few to several dozen days and depends on the species and developmental stage of the tick, the species and physiological status of the host, and external conditions, mainly on temperature. At low temperatures, ticks can be attached to the host for an even longer period (Buczek, Bartosik own field and laboratory studies). 

The role of migratory birds in the spread of tick-borne diseases has long been emphasised. However, although many studies have been conducted in different regions, the biological and physiological relationships between different species of birds, ticks, and pathogens are still not fully elucidated. The increase in the number of tick-borne bacterial diseases emerging in Europe has prompted monitoring of the occurrence and behaviour of not only ticks but also their hosts, given the dynamic changes in the climatic and environmental conditions and, consequently, in the periods of migration and breeding of passerine birds that are carriers of ticks and tick-borne pathogens [28]. Since tick-borne zoonotic diseases are diagnosed in new environments, often distant from previously known locations, comprehensive studies are required to determine the modes of the spread of their etiological factors.

In this review, we analyse the role of passerine birds in the transmission of ticks and pathogens in Europe and highlight the impact of anthropogenic factors on birds’ migratory behaviour and the prevalence of tick-borne zoonoses that are especially important for public health. 

The relevance of this issue is supported by predictions of further expansion of the area of tick occurrence and an increase in the population size of these arthropods in different parts of the world [29,30,31,32,33]. This may result in an increase in the number of tick infestations of hosts, including birds, and expansion of the areas of prevalence of tick-borne diseases.

## 2. Tick Species Most Frequently Infesting Migratory Birds in Europe 

In the non-parasitic phase of their life cycle, ticks reside on the soil surface and in the lower parts of plants, where they find most favourable humidity conditions and have the greatest chance of finding a host. Hence, the highest prevalence of tick infestation is noted in ground-foraging birds, especially species from the orders Passeriformes (Alaudidae and Corvidae) and Galliformes (Phasianidae) [12,34,35,36,37,38,39,40,41]. 

The occurrence of ticks on birds in different regions is determined by the rhythms of diurnal and seasonal activity of individual species and the rhythms of birds’ migrations. During northbound spring migrations of birds over the Baltic Sea, the highest prevalence of tick infestation were noted in *Prunella modularis* (dunnock) (63.1%), *Turdus merula* (common blackbird) (59.1%), and *Turdus iliacus* (redwing) (40.0%). In turn, during migrations to the south between the end of summer and autumn, *Turdus merula* (73.9%), *Fringilla coelebs* (common chaffinch) (64.3%), *Erithacus rubecula* (European robin) (36.2%), and *Turdus philomelos* (song thrush) (35.6%) were found to be the main tick hosts [42].

In Southern Denmark, as many as 43% of 44 different bird species captured during spring and autumn migrations and non-migratory/resident species in different Sub-Saharan, South European, and North European wintering grounds were carriers of ixodid ticks. In South European wintering grounds, tick infestations were detected in *Erithacus rubecula* (60% in spring and approx. 23% in autumn), *Prunella modularis* (16.5% in spring and 0% in autumn), *Sylvia atricapilla* (Eurasian blackcap) (0% in spring and 7.8% in autumn), and *Turdus philomelos* (0% in spring and 16.6% in autumn). Among birds present in North European wintering grounds, ticks were found in *Troglodytes troglodytes* (Eurasian wren) (0% in spring and 16.5% in autumn) and *Turdus merula* (29% in autumn) [41]. Ticks and tick-borne pathogens are transmitted by birds exhibiting different lifestyles, i.e., non-migrants residing in a specific area, short-distance migrants (covering short distances, e.g., from higher mountain areas to those located at lower altitudes), medium-distance migrants (moving within one or several European countries), and long-distance migrants (moving between distant habitats, e.g., breeding sites in Northern Europe and wintering grounds in Central and South Africa).

A higher prevalence of tick infestation was recorded in sedentary birds [40,43].

An important role in local or short-distance transport of ticks is played by common and familiar birds living in urban parks and gardens as well as birds migrating between regions situated at short distances. In a vast area of Central [40,42,44,45,46], Northern and North-eastern [41,47,48], and Southern [49] Europe, a high prevalence of ticks was noted in sedentary *Turdus* birds (e.g., the common blackbirds *Turdus merula*) or short-distance migratory birds (e.g., the song thrush *Turdus philomelos*). In Western Estonia, the highest prevalence of tick parasitism in 24 bird species migrating southwards was noted in representatives of the genus *Acrocephalus* (58%) and, to a lesser extent, in the genera *Turdus* (13%), *Sylvia* (8%), and *Parus* (6%) [50].

In Slovakia, ticks were found in 37.2% of 43 bird species, with the highest infestation rate in the great tit (*Parus major*) (83.8%, 31/37) [51]. These birds usually sit on trees in wooded urban areas and feed on the ground. Northern *P. major* populations migrate slightly southwards before winter (from Poland to France, the Netherlands, and Germany).

The presence and species composition of ticks on birds depends on the migrating and feeding behaviour of the hosts and on the species of ticks occurring in habitats where birds spend the breeding season and the wintering period and/or where they stop during seasonal flights. 

Larvae and nymphs of the common tick *I. ricinus*, which is the most prevalent species in forest and urban areas, are most frequently collected from birds captured in areas north of latitude ca. 58°N [50,52]. In this part of Europe, *I. ricinus* specimens accounted for 91.1% and 92.25% of all ticks collected from migratory birds in the Netherlands and Belgium [53] and in the Danube Delta [45], respectively. An even higher prevalence of this species was recorded on the Baltic Sea coast in Poland (97.5%) [14] and in Western Estonia (99.6%) [50]. In the south of the continent, i.e., Northern Italy, only 8.1% of captured migratory birds were infested by immature *I. ricinus* [54]. In areas below latitude 42°N, such as the Lazio Region in Central Italy *Hy. marginatum* (27.7%), *Hy. marginatum rufipes* (51.8%), *Hyalomma* spp. (12.4%), and rarely *Amblyomma* spp. (3.6%), *I. ricinus* (0.7%), and *Ixodes* spp. (3.6%) were identified most frequently on 41 birds belonging to 17 species during the spring and autumn seasons [55]. On Capri and Antikythira in the European Mediterranean area, only 2.7% of all captured birds migrating northwards in spring were infected by ticks, and *Hy. marginatum* sensu lato was the dominant species. In the collections of Wallménius et al. [13], the species constituted 90% of all bird-infesting ticks. Other taxa found on the Mediterranean islands were *Ixodes frontalis*, *Amblyomma* sp., *Haemaphysalis* sp., *Rhipicephalus* sp., and unidentified ixodids.

Besides *I. ricinus*, other representatives of *Ixodes* were found on migrating birds in various biogeographic regions of Europe. These included: *Ixodes acuminatus* [56], *Ixodes arboricola* [14,42,45,57,58,59] *Ixodes canisuga* [40], *Ixodes eldaricus* [42,60], *Ixodes festai* [61], *I. frontalis* [13,14,27,38,42,44,52,56,61,62,63] *Ixodes hexagonus* [40], *Ixodes lividus* [27,40,64], *Ixodes persulcatus* [50], and *Ixodes redikorzevi* [45,57,58,61] as well as representatives of *Haemaphysalis*, i.e., *Ha. concinna* [27,58,61], *Haemaphysalis punctata* [14,45,61,63], *Haemaphysalis parva* [57], *Haemaphysalis sulcata* [61], and *Hy. marginatum* group [56,57,65,66].

The highest diversity of tick species present on migratory birds and a high prevalence of ticks were recorded in stopover sites along birds’ seasonal migration routes. For instance, along the north-west migration route in Turkey, several tick species from the Ixodidae family: *Ha. concinna*, *Ha. punctata*, *Ha. sulcata*, *Hy. marginatum* group, *I. eldaricus*, *I. festai*, *I. frontalis*, *I. redikorzevi*, and *I. ricinus*, were found on birds [61]. In the area of the intersection of migration routes of various bird species from the south and east to the north in Central Europe, 10 out of 46 species of captured birds were infested by *I. ricinus* (92.25%), *I. arboricola* (6.25%), *I. redikorzevi* (1.00%), and *Ha. punctata* (0.50%); in this group, migratory birds were tick hosts more frequently than resident birds [45].

In stopover sites occupied by birds during seasonal migrations, engorged ticks can detach from their host and colonise a new habitat offering favourable conditions. In these sites, migratory birds can be attacked by ticks, which can then be transported to other places by these hosts.

The distance over which feeding ticks are transported is determined by atmospheric conditions, e.g., temperature, precipitation, and air currents, which can alter birds’ seasonal migration routes. Additionally, at low ambient temperatures, the tick metabolism and feeding dynamics are slowed down (Buczek’s own observations), which promotes longer attachment of ticks to birds; therefore, they can be transported over longer distances.

Birds are most frequently infested by tick larvae and nymphs, but rarely by adult forms [13,39,44,51]. The structure of immature tick populations on birds depends on many factors, but primarily on the biological and physiological traits of the ectoparasites and their hosts, the period and area of research, and the type of habitat where the birds resided before being captured. In Slovakia, larvae and nymphs accounted for 77.6% and 23.4% of all ticks (594) feeding on birds, respectively [51]. On the Baltic coast, larvae and nymphs constituted 54.6% and 46% of 3041 *I. ricinus* specimens, respectively [14]. In turn, nymphs (65.1%) dominated over larvae (32.96%) in reports from Germany [40].

Long-distance migratory birds are more likely to be infested by nymphs than larvae, which may be associated with the duration of feeding of these stages. As indicated by laboratory investigations conducted in the same conditions, larvae of ixodid tick species occurring in Eurasia, e.g., *I. ricinus*, *Ha. concinna*, *Ha. inermis*, *Hy. marginatum*, *D. reticulatus*, and *D. marginatus*, usually ingest blood for a shorter time than nymphs (Buczek, own observations).

The tick fauna on migratory birds captured in Europe differs in the different parts of the continent and at the different times of seasonal migrations. As demonstrated by Poupon et al. [52], birds migrating southwards in Central Europe exhibited a three-fold higher rate of tick infestation than those migrating towards the north, and *I. ricinus* was the dominant species during autumn southward migrations, while *I. frontalis* was found most frequently in birds flying northwards during spring migrations. In spring, birds migrating to the north and captured on the Baltic Sea coast were parasitized by the highest number of *I. ricinus* specimens (169 larvae and 260 nymphs) and *Ixodes arboricola* (112 larvae, 21 nymphs, 1 female), but less frequently by other ticks, i.e., *I. frontalis* (1 larva and 14 nymphs) and *I. eldaricus* (2 females and 2 males). In autumn, only one tick species (*I. ricinus*) was recorded on this migration route [42]. The differences in the species composition of ticks observed between the different seasons of the year were determined by the rhythms of seasonal activity of the individual developmental stages of species, especially the juvenile stages, which attack birds most frequently.

Long-term monitoring of some regions indicated increasing tick prevalence on birds. Along the Baltic Sea coast in Northern Poland, a four-fold increase in the percentage of tick-infested migratory birds in spring (from 9.9% to 40.7%) and an over two-fold increase at the end of summer and autumn (from 17.7% to 40.0%) were reported in a 30-year period [14,42].

## 3. Occurrence of Tick-Borne Pathogens in Ticks Infesting Birds in Europe

Ticks infesting birds with sedentary lifestyles and/or migrating in Europe can be infected with viruses, bacteria, and/or protozoa. Most tick-borne pathogens transmitted by passerine birds are pathogenic to humans. As shown by Alekseev et al. [67], as many as 51.8% of ticks collected from migratory birds in the Kaliningrad enclave (the Baltic Region of Russia) during the spring and autumn in 2000 were infected with pathogens causing human diseases. *Borrelia afzelii*, *Borrelia garinii*, and *Borrelia burgdorferi* sensu stricto (92.9%) were the most prevalent, whereas factors of human monocytic ehrlichiosis (HME) and human granulocytic ehrlichiosis (HGE) were detected less frequently (14%). The role of birds in the transmission of etiological factors of human diseases has been indicated in numerous studies carried out later in Europe. The prevalence of these pathogens in ticks collected from birds varies greatly depending on the species of the vector and the host, collection site, and study period. Recently, Klitgaard et al. [41] demonstrated that 60.9% of ticks feeding on birds in Southern Denmark were PCR-positive for the presence of at least one tick-borne pathogen.

*Borrelia* spirochetes are tick-borne pathogens with the highest prevalence in ticks removed from birds captured in Europe [40,52,53,68,69]. The ecological and physiological relationships between spirochetes of the *B. burgdorferi* sensu lato complex, especially *B. garinii* and *Borrelia valaisiana*, birds, and immature *I. ricinus* ticks [52,63,68] have been investigated most comprehensively.

The prevalence of *Borrelia* spirochetes in bird-infesting ticks varies largely depending on the area (Figure 1), season, and year of study. 

In addition to *Borrelia* spirochetes, immature *I. ricinus* ticks collected from birds usually transmit rickettsia of the genus *Rickettsia*, mainly *Rickettsia helvetica* [37,41,43,44,51,54,58,71,72]. Noteworthy, DNA of *Rickettsia* sp. IXLI1 was detected in 100% of female *Ixodes lividus* ticks collected from the sand martin *Riparia riparia* [64]. Other pathogens, such as *Anaplasma phagocytophilum* [44,71], *Coxiella burnetti* [51], *Candidatus Neoehrlichia mikurensis* [41], TBEV [44,50,73], and *Babesia* spp. [37,47,71,74,75], are identified in bird-infesting ticks less frequently.

Juvenile stages of *I. ricinus* ticks collected from birds can be infected by two and more tick-borne pathogens [41,52,67,71]. As shown by Moutailler et al. [76], co-infections by pathogens and symbionts in *I. ricinus* are a common phenomenon in nature. As in the case of infections by one species, pathogen co-infections are more common in *I. ricinus* nymphs than in larvae infesting migratory birds [41,71]. In Europe, co-infection by various *Borrelia* genospecies has been reported most frequently [41,47,52,71]. Co-infections by various *Borrelia* genospecies with *Rickettsia* species [41,53,54,77] or *Borrelia* with other bacteria, e.g., *Candidatus Neoehrlichia mikurensis* [41,77] or *A. phagocytophilum* [53], have been detected as well. Co-infection by various pathogens, including mixed infections by various *Borrelia* genospecies, was estimated at 20% in *Ixodes* nymphs and 3% in larvae [71].

Birds not only transport pathogen-infected ticks, but also some of them in some areas act as zoonotic reservoirs for pathogens, e.g., Lyme borreliosis agents [44,70,78], *R. helvetica* [43,51], *Rickettsia* spp. [44], *A. phagocytophilum* [44], and *C. burnetti* [51].

Besides *I. ricinus*, other ixodid ticks parasitizing migratory birds transmit pathogens and substantially contribute to the maintenance and circulation of human and animal tick-borne diseases. For example, in Romania, the DNA of various *Rickettsia* species was detected in several tick species, i.e., *Ha. concinna* (*R. monacensis*), *I. arboricola* (*R. helvetica*, *R. massiliae*), and *I. redikorzevi* (*R. helvetica*) [58]. In Northern Spain, *B. turdi* spirochetes were identified in *I. frontalis*, *Ha. punctata*, and *I. ricinus* ticks removed from birds, while the human pathogen *B. valaisiana* was detected in *I. frontalis* and *Ha. punctata* [63].

The literature data cited above indicates an important role of birds in the transmission and maintenance of enzootic cycles of tick-borne pathogens in Europe. The possibility of the rapid spread of tick-borne pathogens with birds quickly moving between different habitats necessitates undertaking intensive activities to develop standards for the prophylaxis and diagnosis of tick-borne diseases.

## 4. Impact of Climatic and Environmental Factors on the Behaviour of Migratory Birds

From North and Central Europe to Africa, birds usually migrate along three routes. The western route through Gibraltar is chosen by most birds flying from the breeding grounds in Great Britain and by many birds from the swampy areas of the Dutch and German Wattenmeer and from the Scandinavian Peninsula. The eastern route leads through the eastern part of Europe from the Baltic States, through Belorussia, Ukraine, Romania, and Bulgaria to Turkey and next over the Bosphorus and the Middle East to Africa. The Central European route (or the Adriatic Flyway), which is a bird migration route from Asia and from the northern, eastern, and central parts of Europe, runs from Siberia westwards to Poland. Next, it is parallel to the eastern migration route through Hungary, over the Balkans and the Adriatic Sea, Southern Italy, Sicily, and Malta to Africa. 

The migratory behaviour of birds is triggered by many interacting exogenous factors, such as the photoperiod, temperature, and habitat food resources as well as endogenous factors (genetic traits, hormones) [79,80,81]. Climate and weather changes influence the length of birds’ stay in wintering grounds [82,83] and the course of the seasonal migrations, e.g., the choice of time, place, and length of stopover necessary to rest and replenish energy reserves [81,84,85,86,87,88]. The migratory behaviour of birds also depends on human activity, which contributes to the urbanization of large areas that used to be birds’ habitats, drainage of habitats, and reduction of the surface of swamp areas and food resources in habitats. Light illuminations may cause disorientation and collisions of birds with tall buildings in cities or with oil rigs in the seas as well as alteration in birds’ flight behaviour [89,90,91,92]. They may also interfere with birds’ circadian rhythms [93]. Artificial light pollution at night exerts a particularly adverse effect on passerine birds that migrate mainly at that time [92]. Chemical air and soil contamination have a directly harmful impact on birds, leading to impairment of many physiological processes. This may result in respiratory malfunction, organ failure, growth retardation, reduced reproductive performance, disturbed egg development and hatching, and abandonment of eggs or chicks by adults. Additionally, these contaminants trigger changes in habitats, thereby reducing food resources for migratory birds [94,95]. Means of land and air transport and even recreational activity of people in birds’ resting and feeding ground may induce stress responses and greater consumption of energy required for migration, survival, and breeding [96].

## 5. Conclusions

Migratory birds covering long distances within a short time and stopping in different habitats on their migration routes are the most important carriers of ticks and pathogens to distant biogeographic regions in the world. They can also contribute to the circulation of viral, bacterial, and protozoan zoonoses with great medical and veterinary importance. The climate and environmental changes affecting the behaviour of birds, i.e., carriers of ticks and tick-borne pathogens, are accompanied by changes in the map of areas characterised by a high risk of tick-borne diseases in Europe. Therefore, the priority task is to monitor the migratory behaviour of birds and the transmission of ticks and tick-borne pathogens by these animals and to monitor changes in the climate and weather conditions. Measures for reduction of the impact of anthropogenic factors on the environment should be taken as well.

## Figures and Tables

**Figure 1 ijerph-17-02117-f001:**
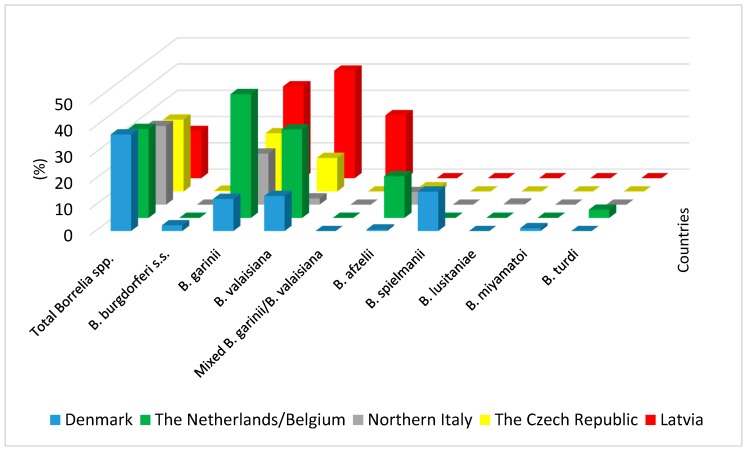
Prevalence of *Borrelia* spirochetes in birds infested by *Ixodes ricinus* ticks in selected areas of Europe [41,53,54,70,71].

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
