# Peer review of "The Potential Role of Migratory Birds in the Rapid Spread of Ticks and Tick-Borne Pathogens in the Changing Climatic and Environmental Conditions in Europe"

_ijerph, 2020, doi:10.3390/ijerph17062117_

Round 1

Reviewer 1 Report

In the Review "The Role of migratory birds in the rapid spread of ticks and tick-borne pathogens in the changing climatic and environmental conditions in Europe" the authors have thoroughly examined the role of migratory birds in the spread of ticks and tick-borne pathogens in Europe.

Below are the comments to improve the manuscript.

The authors should include figures and graphs to summarize their major findings of the review. The scientific names should be denoted in italics uniformly.

Author Response

Dear Reviewers,

We would like to thank the Reviewers for their kind comments on the content of our paper and for the critical remarks and suggestions, which we have addressed in the revised version of the manuscript. All changes made can be traced as they are visible in the revised manuscript.

Review 1

As suggested by the Reviewer, we have presented some data on the occurrence
of Borrelia spirochetes in ticks collected from birds in a graphic form, which has shortened the body of the text. Concurrently, it highlighted the differences in the prevalence of these pathogens in different parts of Europe.

Review 2

We have corrected the errors mentioned by the Reviewer and introduced new information to make the text more understandable to the reader.

Line 29. We wish to explain that we focused in our work on cases of transport
of ticks by birds migrating only within Eurasia and between Eurasia and Africa. Hence, we did not cite publications on tick transport by birds to North America although, as mentioned by the Reviewer, publications on this issue were available as early as in the 1980s.

Review 3

We have introduced the corrections suggested by the Reviewer and explained terms that may not have been fully clear to the reader.

Yours sincerely

Alicja Buczek

Reviewer 2 Report

This manuscript reviews the available literature on tick infestations on birds in Europe. There is a voluminous body of literature on the topic, and a synthesis of that literature in a single publication is worthwhile. I have a few comments.

In general, the role of “changing climatic and environmental conditions” is only superficially discussed in section 5 and is not really related to tick transport by migratory birds. To be consistent with the title and importance, those “conditions” should be better integrated with studies on tick transport. Also, it would be nice to include in the Conclusions a better synthesis of ideas, some actual predictions, and more concrete directions for further study.

Abstract: given the voluminous body of literature and various topics that the review covers, the Abstract seems to be unduly short.

There are many cases throughout the manuscript where Latin names are not italicized.

Line 29: “mites” should be “ticks” because mites are different and not considered here.

Line 59: references are needed at the end of this first statement. The phenomenon of medically-important tick transport by birds has been studied in North America at least since the 1980’s.

Lines 80 through 86: including the families Alaudidae (larks) and Corvidae (crows and jays) and then listing numerous other passerine families is confusing. It is not clear why larks and crows are listed where they are.

Lines 97, 290, and 293: to state that birds are vectors of ticks is somewhat confusing. The ticks are vectors of infectious agents (viruses, bacteria, and protists), while the birds are host of the ticks. I understand that the birds transport the ticks (and even the infectious agents) from one place to another and could be broadly considered as vectors in that sense, but for strictly parasitological semantics, I think that the use of the word “vector” should be confined to the ticks for their role in moving infectious agents from one host to another.

Lines 103 and 104: designations of birds as being sedentary, covering short distances, and being short- or long-distance migrants is vague. Those designations are defined a little bit better from lines 256 to 260. Those definitions should be moved to lines 103 and 104, which is where the designations first appear.

Author Response

(The authors gave the same response as above.)

Reviewer 3 Report

This review attempts to summarize the extent of the literature suggesting a role of birds in the spread of ticks across endemic regions in Europe. However, it is not clear to me what kind of review this is. If it is a systematic or scoping review, there should be a methods section to explain how it was done, i.e. search terms, search platform, how data was trimmed for eligibility. If it is a short communication or an opinion piece, it should be stated as such. However, currently it reads more like a book chapter. It should be rewritten to improve structure and focus.

Additionally, I do not think the extent of the literature on the role of birds in the spread of ticks is currently comprehensive enough to justify this review at this point, even if one would take a global focus on it (birds have been extensively hypothesized to be have a major role in the spread of ticks outside of Europe too). While there is a strong hypothesis to be made for it, it is still too speculative, because most studies describe tick prevalence in birds, but in no way track movement of particular tick individuals and associate it with bird movement, in major part due to the difficulty of the task. One way I could see this manuscript become publishable is if it is refocused into an opinion piece, where all sections would be shortened and all cited studies would be summarized succinctly. I would modify the Introduction and Conclusion to clearly state that it is a hypothesis that currently lacks clear evidence, and should be investigated further (for example, by using molecular approaches, such as using genomic markers to correlate gene flow patterns between birds and their ectoparasites, or isotopic markers to trace origins of birds and their ectoparasites).

Lastly, latin names should be italicized, and there are underscores in the text where there should only be spaces.

I have made a number of specific comments, mostly on the Introduction, to help the authors restructure their manuscript.

Abstract

Line 10 : The first sentence is a little bit unclear. What do you mean by “frequently distant”?

Line 11: The second sentence has the same problem. What do you mean with “previously unknown”?

Introduction

Line 23: The word tick should be singular here.

Line 32: Change “The vector competences” to “Vector competence”. Also, vector competence is discussed more in the following paragraph, so this section should be moved there perhaps.

Line 33: Switch the words “probably” and “they”. This sentence needs citations.

Line 34: The last sentence of this paragraph belongs to the next paragraph.

Line 37: I think this sentence should be rewritten to be more intuitively coherent for the reader. A tick primarily extracts blood meals from their hosts, and saliva dejection is more of a secondary outcome that unfortunately leads to transmission of diseases to hosts, but it is not a primary function of blood feeding. Also, blood feeding does not only happen this way when performed on birds, but rather on all potential hosts, including mammals.

Line 39: I would use the word “ticks” instead of “specimens”. I would also rephrase the last part of the sentence to explain that infected ticks may transmit pathogens in the host’s blood vessel that may be picked by other ticks in the vicinity on the host’s body that are feeding at the same time.

Line 40: I do not understand the sentence starting at this line. It should be rewritten.

Line 44: What do you mean with “move horizontally”? Did you mean that they may disperse on their own over short distances? I do not know the term “Ambushing ticks”. I suggest removing it or explaining it further.

Line 47: This sentence needs citations, and should be expanded more. What are favourable conditions, and what avian and mammalian hosts may transport ticks? How different is movement by means of birds versus mammals?

Line 49: This paragraph should be rewritten to clearly explain that each tick stage feeds one time, and most species in Europe will feed on three different types of host sequentially, but some species will feed on only two, meaning two stages will feed on the same type of host. Also, explain what you mean by type of host. For example, are birds a type of host on which ticks may feed that is different from mammals? Overall, this paragraph would benefit from being more specific with the terms used.

Line 57: This paragraph seems disconnected from the rest of the text, and is very small. Many references and concepts that would be important to the manuscript seem to be missing. For example, how are ticks transported? Is it through hitchhiking while parasitizing birds? When would they be transported? There is a large variety of birds that may be parasitized by a wide variety of tick species. Migration is seasonal, and will be largely dependent on the bird species involved, and also sex and age. Many bird species/populations are permanent resident, which will affect the degree of spread, which may only be confined to the home range of these birds. Ticks can be moved by mammals too. It was briefly mentioned elsewhere in the Introduction, but I think it should be expanded more. These are all aspects that should be emphasized in the Introduction to a much greater extent.

Line 79: The correct spelling is “prevalence”.

Second section

This section mostly lists studies on the prevalence of tick parasitism in birds in various regions of Europe, but does not summarize the extend of the literature on the role of birds on movement between endemic areas. I suggest modifying the title for the section as such, and adding a paragraph to summarize findings from all these studies.

Third section

This section does not seem distinct enough from the previous section to be separated from it. None of these two sections properly questions the role of birds in the movement of ticks. They mostly describe aspects of tick parasitism on birds captured across Europe. I suggest grouping them into one section, and summarizing all the findings in the context of hitchhiking on birds and movement between endemic areas.

Fourth section

Similar to the previous two sections, this section lists studies on the prevalence of pathogens in ticks collected from captured birds in various regions of Europe, but does not summarize the extend of the literature on the role of birds on the transmission cycle of pathogens and on the propagation of pathogens across endemic regions. I suggest modifying the title for the section as such, and adding a paragraph to summarize findings from all these studies.

Author Response

(The authors gave the same response as above.)

Round 2

Reviewer 3 Report

It appears the authors correctly addressed a number of my comments.

However, one of the main issues I had with this opinion piece is that it doesn’t highlight how little evidence we possess about the actual role of birds in propagating ticks. Parasitism prevalence is not equivalent and should never be confused with tick invasion and establishment. For example, ticks hitchhiking on a bird in migration could find itself in an unsuitable habitat on arrival for its survival or reproduction (often seen in Canada), and/or could be incapable of successfully reproducing with local ticks. While there is a large breadth of evidence available to suggest that migratory bird show various levels of tick parasitism, possible across the world, there are very few that actually investigate the role of birds in moving ticks from one area to another and successfully contributing to invasion and establishment of these ticks in new areas, or contributing to gene flow in tick populations across areas frequented by migratory birds. To my knowledge, only studies on molecular ecology using genetic markers or radioactive isotopes could correctly address this problematic, and these studies are currently very scarce. It would be interesting to refer to such studies and explain how our knowledge of the role of migratory birds on tick invasion and establishment could be improved with additional similar studies. The lack of understanding of these concepts by the authors is evidenced by the title and the conclusion section of their opinion piece, which have not been modified sufficiently, if at all, since last version.

Also, another important way ticks can be moved around the landscape is by the means of mammals, and this aspect was not explored sufficiently by the authors, apart from one sentence in the Introduction. This leads me to think that the opinion piece focuses on one mechanism, while mostly disregarding alternative ones, to explain observed trends.

The Introduction also is not structured cohesively to bring the subject in a way that would make the piece interesting for the readership of the journal. I made a number of comments on it, but not all were addressed by the authors.

I will be willing to revised a new version of this manuscript if these major points are addressed in a comprehensive way. If the authors can address the points in a way that is different from the one I suggested, I will be willing to review the manuscript again, but I will need a rebuttal letter addressing each comment with which they disagree.

Author Response

9th March 2010

Dear Reviewer 3,

Once again, thank you for the thorough and kind revision of our manuscript. We have included most of the Reviewer’ comments in the new version. We appreciate them, as they immensely contributed to improvement of the quality of the paper. At the request of the Reviewer 3, we respond in detail to all comments:

  1. 10.

We agree that the use of the word "frequently" was unfortunate. We intended to emphasize that birds play an important role in the transmission of ticks not only in one or closely located habitats but also between remote regions. Foraging of hard ticks (Acari: Ixodida: Ixodidae) is a long-term process. As shown by the literature data and our multiyear investigations, depending on e.g. the tick species and developmental stage, the immunological status of the host, and environmental conditions, it may last from several to even several tens of days. That is why in the revised abstract (l. 11-12), we stated more precisely that birds with different lifestyles are involved in the transmission of ticks

 “Birds with different lifestyles, i.e. non-migrants residing in a specific area, or short-, medium-, and long-distance migrants, migrating within one or several distant geographical regions”

l.11:

We have considered this remark. In l. 14-16, we explain that ticks and pathogens can be transmitted by birds to areas where they did not occur earlier.

l.23.

We have considered this remark. Ticks were changed in Ixodid tick.

  1. 32:

We have followed this suggestion from Reviewer 3 and removed ”the”

l.33. The words probably and they were switched according to the suggestion. There are many literature reports on the competence of the six tick species to transmit different species of pathogens and their contribution to pathogen circulation in nature and in maintenance of tick-borne disease foci (individual tick species have different vector competence or do not have it at all for different species of pathogens; also, their contribution to the maintenance of foci of various diseases may vary). To support this sentence reliably by citing original papers, we would have to include several other publications. This would significantly increase the volume of the paper, while the Reviewers suggested that we should shorten it.

l.34:

We have changed the text in accordance with the suggestion from Reviewer 3 and shifted the sentence to the next paragraph.

l.37 and l.39:

We've addressed the comment. As suggested by the Reviewer, we have improved the sentence so that its content can be understood by readers l. 43-47:

“Tick feeding which consists in introduction of saliva into the host organism alternately with blood meal uptake, may lead to transmission of pathogens in both directions: from the tick to the host and vice versa. Infected ticks may transmit pathogens in the host’s blood vessel that may be picked by other ticks feeding in the vicinity on the same host’s body at the same time [15]”.

l.40.

we have considered the comment to l. 47-49 in the revised version:

“Transfer of certain pathogenic and non-pathogenic microorganisms can also take place via conspecific and interspecific tick parasitism [16] or, probably, during oral-anal contact between two different tick species - I. ricinus and D. reticulatus [17].

l.44.

We have followed the comments. Ticks can move on plants vertically and horizontally. We have corrected the sentence to make it easier for the reader to understand. We have given up the use of the word "ambushing", which is a typical specialist term used by Ixodologists. The term "ambushing ticks" refers to ticks (e.g., Ixodes ricinus, Ixodes scapularis, Dermacentor reticulatus) that wait on plants for their host, as opposed to "hunting ticks" (e.g., Hyalomma marginatum, which follow towards stimuli originating from the host. The revised version is as follow (now lines 50-52):

“In natural conditions, ticks move over short distances. Ixodes scapularis nymphs and adults, cover a distance of only 2-3 m and 5 m, respectively [18]. Within 7 weeks, adult D. reticulatus ticks can cover an average distance of 60.71 ± 44 cm [19]”.

l.47:

We apologize for not having cited papers on favourable conditions in which ticks can colonize new habitats. These may include various abiotic (e.g. temperature, humidity) and biotic factors (e.g. presence of hosts for various stages of tick development). Migration routes and host locations are also affected by various factors, e.g. atmospheric and habitat conditions changing together with climate change, weather phenomena, and anthropopressure. Citing these papers would significantly increase the size of the paper, and other Reviewers mentioned the appropriate length of the manuscript.

l.49.

We have considered the comment and changed the sentence to make it clearer (now lines 65-67):

“Each hard tick stage feeds one time, and most species in Europe feed on three different hosts sequentially, but some species feed only on two hosts as two developmental stages will feed on the same host”.

In accordance with the reviewer's comments, we have supplemented the literature adequate to the content of the paragraph, describing the participation of animals, including mammals, in the transmission of ticks to new territories, [new references 20-23] (now lines 52-53):

:

  1. Dutkiewicz, J.; Siuda, K. Rhipicephalus rossicus Jakimov et Kohl-Jakimova, 1911 - a new tick genus and species (Acarina, Ixodidae) in the Polish fauna. Faun. 1969, 15, 99-105 (in Polish).
  2. Qviller, L.; Risnes-Olsen, N.; Bærum, K.M.; Meisingset, E.L.; Loe, L.E.; Ytrehus, B.; Viljugrein, H.; Mysterud, A. Landscape level variation in tick abundance relative to seasonal migration in red deer. PLoS One. 2013, 8, e71299. doi: 10.1371/journal.pone.0071299.
  3. Mysterud, A.; Qviller, L.; Meisingset, E.L.; Viljugrein, H. Parasite load and seasonal migration in red deer. 2016, 180, 401–407. doi: 10.1007/s00442-015-3465-5.
  4. Vial, L.; Stachurski, F.; Leblond, A.; Huber, K.; Vourc'h, G.; René-Martellet, M.; Desjardins, I.; Balança, G.; Grosbois, V.; Pradier, S.; Gély, M.; Appelgren, A.; Estrada-Peña, A. Strong evidence for the presence of the tick Hyalomma marginatum Koch, 1844 in southern continental France. Ticks Tick Borne Dis. 2016, 7, 1162-1167. doi: 10.1016/j.ttbdis.2016.08.002.

We also added an explanation as follow (lines 52-60):

“These hosts migrate regularly (cyclically), which is related to e.g. reproduction cycles and recurrent changes in the environment. In turn, their irregular migrations are most often related to adverse environmental conditions (e.g. lack of feed or water) and overpopulation. Ticks infesting the skin their avian and mammalian hosts are spread within and among habitats. Ticks attached to mammalian fur can also be transferred over certain distances. We found labelled unengorged adult D. reticulatus ticks at a distance of 2 to 3 km away from the site where they were released. They were probably transferred on mammalian fur or on the clothes of forest workers that were present in the habitats of these ticks [19, Bzowski, unpublished data]”.

We agree with the Reviewer's remark that the importance of migrating birds in determination of the species structure of ticks in a given area can be assessed only after conducting other investigations, also genetic studies. We have added a sentence in l. 61-64 to draw readers' attention to the problem:

 “However, determination of the impact of birds on the transmission and fauna of ticks in various regions requires further research, including investigations of molecular ecology with genetic methods based on genetic markers or radioactive isotopes”

To our knowledge, no studies have been carried out on the molecular ecology of ticks transmitted by migratory birds in Europe (we have considered this area). We also did not find this type of research in the world literature.

In acknowledging the Reviewer's point of view, we have added word “potential” in title.

l.57.  

As suggested by the Reviewer 3 we have added information on tick transfer (now line 72-77):

“Ticks attached to host skin are transmitted from one habitat to another mainly by avian and mammalian hosts. The length of tick foraging varies from a few to several dozen days and depends on the species and developmental stage of the tick, the species and physiological status of the host, and external conditions, mainly on temperature. At low temperatures, ticks can be attached to the host for an even longer period (Buczek, Bartosik own field and laboratory studies). In favourable conditions, ticks can colonise a new habitat and reproduce successfully“.

There is no suggestion in the title of the manuscript about a role of birds other than transmission of ticks to other habitats. When writing about the "fast" transmission route, we only underlined that birds can carry ticks over long distances within a very short time (as long-distance migrants can travel many kilometers a day).

In response to the comment, we have expanded the revised version with the information that Ixodid ticks feed from a few to several dozen days, depending on the species, developmental stage, and other factors (e.g. environmental conditions, host physiological characteristics). Our research clearly confirms that long feeding of ticks on hosts enables them to be transferred to new areas. In laboratory conditions, we reared ticks representing various genera, e.g. Ixodes, Haemaphysalis, Hyalomma, Rhipicephalus, and Dermacentor; hence,  we have precise data on the length of feeding in larvae, nymphs, and adult forms of various tick species. What happens to transmitted ticks in new habitats is a different matter.

Second section: “Spread of ticks between endemic areas of tick-borne diseases by birds” and third sections “Tick species most frequently infesting migratory birds in Europe” have been fused (now line 95)

With kind regards and thanks for your support

Authors

Round 3

Reviewer 3 Report

Although I did not see the last version of the manuscript (the v2 uploaded here seemed to be an unmodified one), I read all the authors' responses and it appears that they modified their manuscript substantially to address my comments. Therefore, I recommend publication. Looking forward to see the final version online.